METHODS AND RESOURCES

# *Aequorea*'s secrets revealed: New fluorescent proteins with unique properties for bioimaging and biosensing

Gerard G. Lambert[1], Hadrien Depernet[2], Guillaume Gotthard[2], Darrin T. Schultz[3,4], Isabelle Navizet[5], Talley Lambert[6,7], Stephen R. Adams[8], Albertina Torreblanca-Zanca[1], Meihua Chu[1], Daphne S. Bindels[9], Vincent Levesque[10], Jennifer Nero Moffatt[10], Anya Salih[11], Antoine Royant[2,12], Nathan C. Shaner[1] *

**1** Department of Neurosciences, Center for Research in Biological Systems, University of California San Diego School of Medicine, La Jolla, California, United States of America, **2** Structural Biology Group, European Synchrotron Radiation Facility, Grenoble, France, **3** University of California Santa Cruz, Santa Cruz, California, United States of America, **4** Monterey Bay Aquarium Research Institute, Moss Landing, California, United States of America, **5** Laboratoire Modélisation et Simulation Multi-Echelle, Université Gustave Eiffel, Université Paris Est Creteil, Marne-la-Vallée, France, **6** Department of Cell Biology, Harvard Medical School, Boston, Massachusetts, United States of America, **7** Department of Systems Biology, Harvard Medical School, Boston, Massachusetts, United States of America, **8** Department of Pharmacology, University of California San Diego School of Medicine, La Jolla, California, United States of America, **9** Nikon Imaging Center, University of California San Diego, La Jolla, California, United States of America, **10** Birch Aquarium at Scripps, La Jolla, California, United States of America, **11** Confocal Facility, Western Sydney University, Penrith, New South Wales, Australia, **12** Institut de Biologie Structurale, Université Grenoble Alpes, CNRS, CEA, Grenoble, France

* ncshaner@ucsd.edu

**Data Availability Statement:** A large portion of the relevant data are within the paper and its Supporting Information files. The native cDNA sequences for the coding region of each FP

## Abstract

Using mRNA sequencing and de novo transcriptome assembly, we identified, cloned, and characterized 9 previously undiscovered fluorescent protein (FP) homologs from *Aequorea victoria* and a related *Aequorea* species, with most sequences highly divergent from *A. victoria* green fluorescent protein (avGFP). Among these FPs are the brightest green fluorescent protein (GFP) homolog yet characterized and a reversibly photochromic FP that responds to UV and blue light. Beyond green emitters, *Aequorea* species express purple- and blue-pigmented chromoproteins (CPs) with absorbances ranging from green to far-red, including 2 that are photoconvertible. X-ray crystallography revealed that *Aequorea* CPs contain a chemically novel chromophore with an unexpected crosslink to the main polypeptide chain. Because of the unique attributes of several of these newly discovered FPs, we expect that *Aequorea* will, once again, give rise to an entirely new generation of useful probes for bioimaging and biosensing.

## Introduction

EGFP and other engineered variants of avGFP [1] have truly transformed biological imaging, allowing researchers to probe living cells in ways that were previously unthinkable [2–4]. The seemingly impossible task of producing a bright avGFP variant with an emission peak beyond

transcript described here have been deposited in GenBank, accession numbers MN114103 through MN114112. Raw Illumina RNA-Seq reads have been deposited in the NCBI Sequence Read Archive (SRA), accession numbers SRR9606756 through SRR9606760. Plasmids encoding the FPs described in this manuscript have been deposited with AddGene (plasmid numbers 129499 through 129512). The structures of AausFP1 and AausFP2 have been deposited in the Protein Data Bank under entry codes 6S67 and 6S68, respectively. Spectra from Fig 2 and photophysical characterization data from Table 1 are available on FPbase.org. Raw image data from all microscopy experiments are accessible from the DOI https://doi.org/10.26300/4x48-y393.

**Funding:** This work was supported by the following grant awards: NIH R01GM109984 (GGL, ATZ, MC, DSB, and NCS), NIH R01GM121944 (GGL, ATZ, MC, DSB, and NCS), NIH U01NS099709 (GGL, ATZ, MC, DSB, and NCS), NIH R21EY030716 (GGL, ATZ, MC, DSB, and NCS), NIH U01NS113294 (GGL, ATZ, MC, DSB, and NCS), NSF NeuroNex 1707352 (NCS), and NIH R01GM086197 (SRA). HD was supported by a PhD fellowship from the European Synchrotron Research Facility (https://www.esrf.eu). The funders had no role in study design, data collection and analysis, decision to publish, or preparation of the manuscript. GGL, ATZ, MC, DSB, and NCS received salary support from the funding sources listed above.

**Competing interests:** The authors have declared that no competing interests exist.

**Abbreviations:** avGFP, *Aequorea victoria* green fluorescent protein; CP, chromoprotein; FP, fluorescent protein; FRET, Förster resonance energy transfer; GFP, green fluorescent protein; H2B, histone 2B; mRNA-Seq, mRNA sequencing; OSER, organized smooth endoplasmic reticulum.

yellow-green [5] drove many groups to explore other marine organisms as potential sources of fluorescent proteins (FPs) emitting at longer wavelengths. About 5 years after avGFP was cloned, FPs were discovered in corals [6,7], and since that time, FPs cloned from jellies, corals, and many other marine organisms have been reported (e.g., [8–10], among many others).

Despite this abundance of reported wild-type FPs, most FPs in widespread use as imaging tools are derived from only a handful of these organisms. Numerous avGFP variants with blue, cyan, green, and yellow-green emission remain the workhorses of live-cell imaging, and derivatives of red-emitting FPs from the soft coral *Discosoma* sp. [6,11,12] and the sea anemone *Entacmaea quadricolor* [10,13–15] make up the majority of commonly used FPs emitting at longer wavelengths. With the practical limitations of these particular FP scaffolds becoming more apparent as live-cell microscopy grows more complex and demanding, our group has focused on identifying, characterizing, and engineering FPs with low homology to these traditional choices. Our ongoing efforts include cloning new FPs from diverse sources as well as investigating previously un- or under-characterized FPs to identify new scaffolds with improved and/or novel spectral properties from which to launch new engineering efforts.

While searching for organisms expressing new and unusual FPs at Heron Island, a research station in the southern Great Barrier Reef, we collected a single individual of an unknown *Aequorea* species that we later determined was most similar to *A. australis*. This serendipitous encounter with a familiar genus led us to discover several novel FP homologs from 2 *Aequorea* species. Several of these proteins offer unique starting points for probe engineering.

## Results and Discussion

The cyan-blue coloration of the radial canals of the *A*. cf. *australis* specimen we collected (Figs 1A, S1, and S2, and Figs K–N in S1 Text) suggested the potential presence of red-absorbing chromoproteins (CPs) and led us to reconstruct the transcriptome of the animal. In addition to transcripts encoding an FP clearly homologous to *A. victoria* green fluorescent protein (avGFP), as we expected, the *A*. cf. *australis* transcriptome also contains several transcripts encoding other, much more divergent avGFP homologs. Characterization of the recombinant proteins revealed that these avGFP homologs include a green-emitting FP that appears to be the brightest FP discovered to date, 2 long-wavelength-absorbing CPs, and a reversibly photochromic FP (Figs 1B and 2; Table 1).

Intrigued by the diversity of optical properties in the *A*. cf. *australis* FP homologs, we next investigated a sample of *A. victoria* from the Crystal Jelly exhibit at the Birch Aquarium at Scripps to determine whether this species also contained multiple diverse FPs. As we suspected, the *A. victoria* individual we sequenced expressed avGFP as well as orthologs of the bright green-emitting FP and the unusual CPs that we first identified in *A*. cf. *australis*. Surprisingly, this *A. victoria* also expressed a fairly close homolog of avGFP with surprisingly EGFP-like properties: a fully anionic chromophore, low $pK_a$, and much more efficient folding and maturation at 37°C than wild-type avGFP. We found that only 2 mutations—1 to speed maturation at 37°C and 1 to monomerize the protein—generated an FP with properties comparable to the commonly used avGFP variant mEGFP. *A. victoria*'s 2 CPs are distinct from those of *A*. cf. *australis*, undergoing an apparently unique mode of photoconversion from a green-emitting (green fluorescent protein [GFP]–like) state to a non-fluorescent, orange-red-absorbing state after exposure to blue light (Fig 2). While not characterized in depth during this study, this unusual property certainly warrants additional investigation of these CPs.

The 3 broad classes of FP homolog found in this study—green-emitting FPs, long-wavelength-absorbing CPs, and a reversibly photoswitchable CP—are discussed below. A phylogenetic tree of the FP homologs from this study is shown in Fig 3, and a sequence alignment is shown in Fig A in S1 Text.

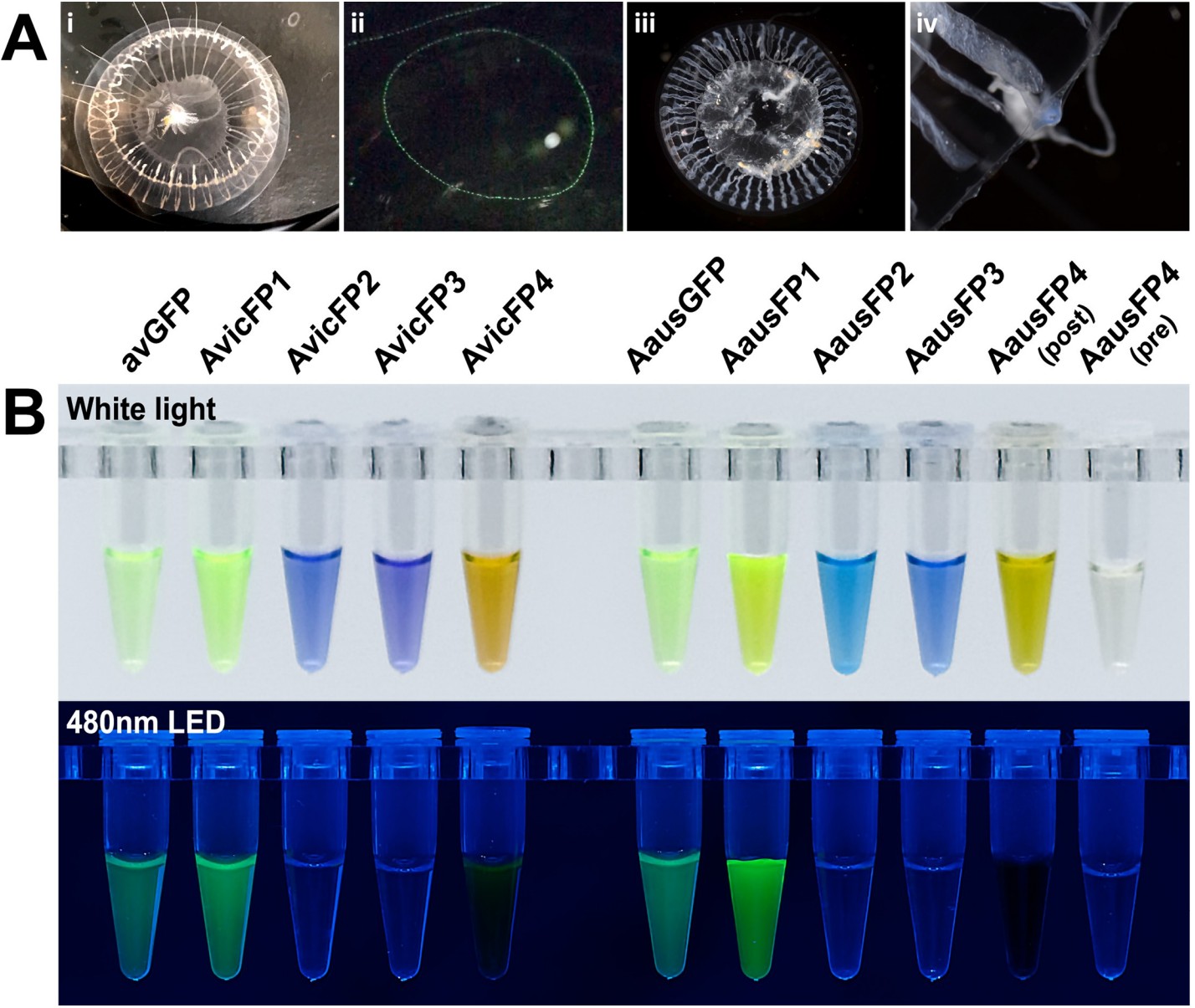

**Fig 1. Photographs of *Aequorea* individuals from this study and purified fluorescent proteins cloned from these samples.** (A) White-light (i) and fluorescence (400-nm LED illumination) (ii) photographs of *A. victoria* and white-light photographs of *A.* cf. *australis* (iii, iv). The blue coloration of *A.* cf. *australis* is shown in the higher magnification image of one of its tentacle bulbs (iv). (B) Purified recombinant proteins from *Aequorea* species, shown under white light and 480-nm LED without emission filters. Protein concentrations were adjusted to display similar optical density as judged by eye and were between 0.5 and 2 mg/ml for all samples.

## Multiple, diverse *Aequorea* GFPs

As expected, both *Aequorea* species abundantly express close homologs of avGFP. Both AausGFP (*A.* cf. *australis* GFP) and the avGFP sequence identified in this work possess optical and biochemical properties similar to Prasher et al.'s original avGFP clone [1], characterized by an excitation spectrum with 2 peaks at 398 and 477 nm and an emission peak at 503 nm. The extinction coefficient and quantum yield of the 2 proteins are similar, and in our hands match closely with the literature values for avGFP [2].

We were surprised to discover a second green-emitting FP in *A.* cf. *australis*, AausFP1, that shares only 53% amino acid identity with avGFP (Fig A in S1 Text). AausFP1 is to our

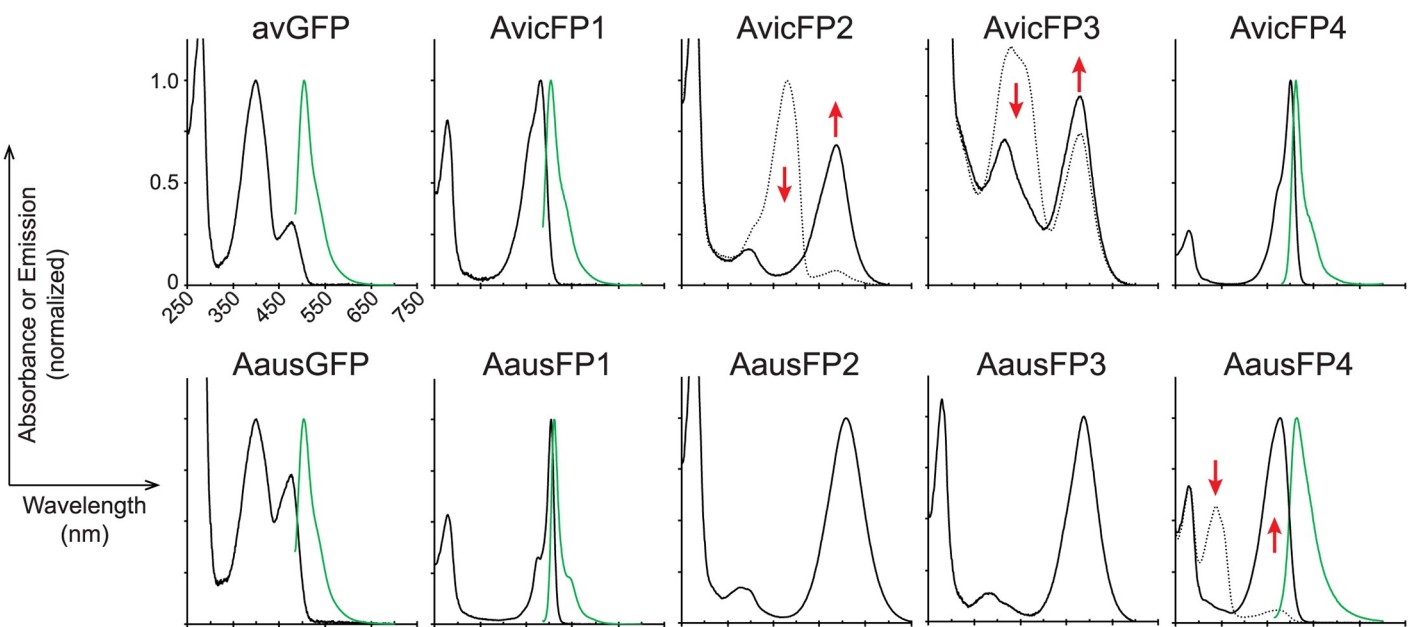

**Fig 2. Absorbance and emission spectra (where measurable) for FP homologs in this study.** Proteins from each species were designated AvicFP or AausFP and numbered in order of discovery, with chromoproteins retaining the "FP" nomenclature for consistency. For photoswitchable and photoconvertible proteins, pre-illumination absorbance spectra are shown as dotted lines, and post-illumination absorbance spectra as solid lines. Emission spectra are shown as green solid lines. The emission spectra for AvicFP2 and AvicFP3 were measured using 460-nm excitation prior to photoconversion. The emission spectrum of AausFP4 was measured using 440-nm excitation after photoswitching to the blue-absorbing state. Red arrows indicate peaks that increase or decrease upon photoconversion or switching. For ease of display, spectra are normalized to the maximum visible absorbance for non-photoactive proteins, and to the pre- (for AvicFP2) or post-illumination (for AvicFP3 and AausFP4) maximum for photoactive proteins. All plots share the same *x*-axis scale as shown for AausGFP. The data underlying this figure may be found at FPbase (https://www.fpbase.org). FP, fluorescent protein.

knowledge the brightest FP discovered to date, with a nearly perfect quantum yield (0.97) and a peak extinction coefficient of 170,000 $M^{-1}cm^{-1}$, making it nearly 5-fold brighter than EGFP on a per-molecule basis. These already extraordinary properties are further bolstered by a low fluorescence $pK_a$ (4.4), unusually narrow excitation and emission peaks (see Fig 2; the emission peak of AausFP1 has a full width at half maximum [FWHM] of 19 nm, compared to 32 nm for EGFP), and higher photostability than mEGFP (see below). The ortholog of AausFP1 in *A. victoria*, AvicFP4, shares some of its unusual properties, such as narrow excitation and emission peaks (Fig 2), efficient folding at 37°C, and a fairly high extinction coefficient, but its low quantum yield (0.10) makes it the dimmest GFP found in *A. victoria*.

AausFP1 was expressed at very low levels relative to other FPs in the *A.* cf. *australis* individual sequenced (see Table A in S1 Text) and would be rare or absent in most cDNA expression-cloning libraries. The transcriptomic approach used in this study is the only practical way to identify such unusual, low-abundance FPs, short of costly whole genome sequencing. Despite low expression in its native context, wild-type AausFP1 expresses and folds very efficiently in *E. coli* at 37°C without any modifications. Though brightly fluorescent, AausFP1 is largely insoluble in this context, and when purified, the soluble fraction of the protein runs as a high-molecular-weight aggregate on size exclusion chromatography (Fig BB in S1 Text).

When expressed in mammalian cells, AausFP1 is excluded from the nucleus and only forms visible aggregates in the most highly expressing cells (Fig W in S1 Text), suggesting that it may form soluble but high-molecular-weight aggregates in this context as well. X-ray crystallography revealed a uniquely stabilized chromophore environment in AausFP1 that may be responsible for its unique properties (see Fig 4, Tables C–E in S1 Text, and Figs B, D, E, and G in S1 Text). Since AausFP1 crystallizes as a dimer, we speculate that it takes on this oligomeric

**Table 1. Photophysical properties of fluorescent proteins described in this study derived from *A. victoria* and *A.* cf. *australis*.**

| Protein | $\lambda_{abs}$ [a] | $\lambda_{em}$ [b] | $\varepsilon$ [c] | $\varphi$ [d] | Brightness[e] | $pK_a$ [f] | Photostability[g] |
|---|---|---|---|---|---|---|---|
| avGFP[h] | 398/477[i] | 503 | 41 (6.5)/15 (2.2) | 0.75 (0.01) | 91/32 | 4.8[j] (0.1) | ND |
| AvicFP1 | 481 | 503 | 64 (3.7) | 0.63 (0.03) | 118 | 4.9 (0.1) | ND |
| AvicFP2[k] | 480 | 515 | 59 (1.2) | 0.04 (0.01) | 6 | ND[m] | ND |
| | 588 | —[l] | 41 (3.2) | — | — | | |
| AvicFP3[k] | 480 | 520 | ND | <0.001[l] | — | ND | ND |
| | 580 | — | ND | — | — | | |
| AvicFP4 | 500 | 512 | 121 (2.1) | 0.10 (0.01) | 36 | ND | ND |
| AausGFP[m] | 398/477 | 503 | 29 (1.6)/22 (1.3) | 0.73 (0.01) | 62/47 | 4.8[i] (0.1) | ND |
| AausFP1 | 504 | 510 | 170 (6.0) | 0.97 (0.05) | 485 | 4.4 (0.1) | 129 ± 4 (*N* = 20) 218 ± 9 (*N* = 23) |
| AausFP2 | 609 | — | 52 (2.3) | — | — | <6.0 | ND |
| AausFP3 | 587 | — | 59 (1.7) | — | — | <6.5 | ND |
| AausFP4[n] | 338/477 | —/510 | 42 (1.9)/3 (0.1) | —/<0.001 | — | ND | ND |
| | 477 | 513 | 69 (2.4) | <0.001 | | | |
| mAvicFP1 | 480 | 503 | 65 (0.04) | 0.63 (0.01) | 126 | 4.9 (0.1) | 131 ± 3 (*N* = 20) 121 ± 8 (*N* = 20) |
| mEGFP[o] | 488 | 507 | 56 | 0.60 | 100 | 6.0 | 100 ± 4 (*N* = 31) 100 ± 4 (*N* = 13) |
| mNeonGreen[p] | 506 | 517 | 116 | 0.80 | 274 | 5.7 | ND |

The commonly used protein mEGFP and the bright monomeric FP mNeonGreen are included for comparison. Values reported for photophysical parameters are the mean of at least 3 independent measurements on independently prepared samples; values in parentheses are standard deviation of the mean unless otherwise noted below. A dash indicates no measurable emission or negligible brightness.

[a]Peak absorbance wavelength (nm).

[b]Peak emission wavelength (nm).

[c]Molar extinction coefficient ($mM^{-1}cm^{-1}$).

[d]Fluorescence quantum yield.

[e]Brightness ($\varepsilon \times \varphi$), percent normalized to mEGFP.

[f]For FPs with a quantum yield $\geq 0.10$, the reported $pK_a$ is the pH at which fluorescence emission is 50% of maximal brightness; for AausFP2 and AausFP3, the $pK_a$ was determined only approximately and represents the pH at which the long-wavelength absorbance peak is 50% of its maximal value.

[g]Mean photobleaching half-times in live cells, corrected for molecular brightness and scaled to the half-time measured for mEGFP in this study (mEGFP = 100%); half-times under widefield (upper value) or laser scanning confocal (lower value) illumination (see Methods and S1 Text) are shown ± the standard error of the mean, with the number of individual cells sampled given in parentheses.

[h]Measured in this study using the avGFP peptide sequence from the *A. victoria* individual sequenced.

[i]avGFP displays two absorbance peaks whose ratio is largely insensitive to pH changes over much of the physiological range but is somewhat sensitive to protein concentration; values separated by slashes in all columns represent those for these two distinct peaks, respectively.

[j]Fluorescence $pK_a$ value determined by exciting the 477-nm absorbance peak.

[k]Values given are for pre- (upper value) and post-photoconverted (lower value) forms of AvicFP2 and AvicFP3.

[l]Fluorescence quantum yields less than 0.001 were not determined, even in cases with a measurable emission peak.

[m]AausGFP is the closest direct homolog to avGFP from *A.* cf. *australis* and displays a similar double-peaked absorbance; values separated by slashes represent those from each peak, respectively.

[n]AausFP4 is reversibly photoswitchable between a UV-absorbing form and a blue-absorbing form; the UV-absorbing form has a small amount of residual blue absorbance; values of photophysical parameters UV and blue absorbance peaks are separated by slashes.

[o]Values from [2] are shown in the table and were re-verified in this study.

[p]Values from [35] are shown in the table and were re-verified in this study.

avGFP, *Aequorea victoria* green fluorescent protein; FP, fluorescent protein; ND, not determined.

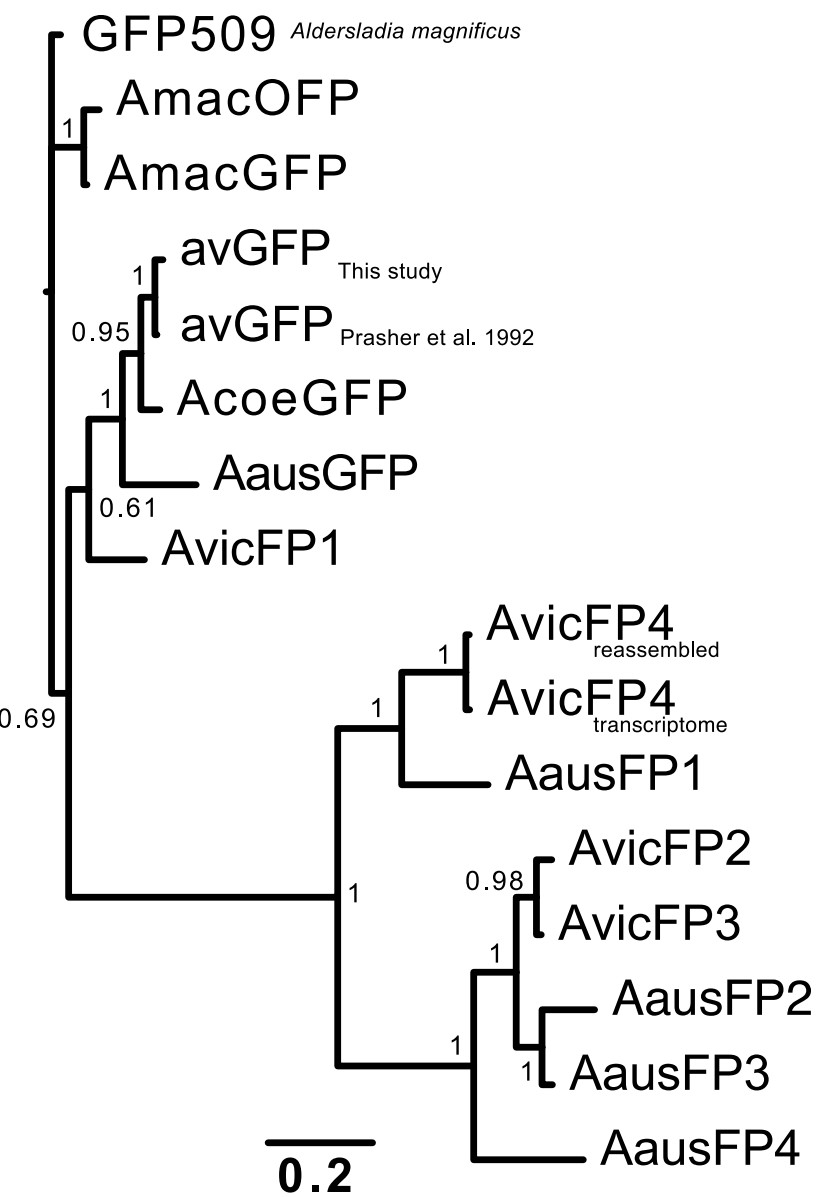

**Fig 3. Phylogenetic tree for FPs cloned in this study, with *Aequorea macrodactyla* and *Aldersladia magnificus* green FPs included as outgroups.** Green-emitting FPs with avGFP-like properties, including AvicFP1, fall into 1 cluster of fairly closely related sequences, while the novel fluorescent (AausFP1 and AvicFP4) and non-fluorescent homologs form 2 additional families. The data underlying this figure (nucleotide sequences of the FPs from this study) may be found in GenBank, accession numbers MN114103 through MN114112. avGFP, *Aequorea victoria* green fluorescent protein; FP, fluorescent protein.

state in its native context, perhaps stabilized by other interactions. It is possible that, as with other FPs such as dTomato [11], the dimer interface serves to stabilize the chromophore environment and may contribute to the unusually high brightness of AausFP1.

AausFP1 photobleaches at similar rates to mEGFP on both widefield and confocal microscopy when instrument settings are identical, but because AausFP1 emits photons at a higher rate (due to its high quantum yield and extinction coefficient), its true photostability is somewhat higher than that of mEGFP (S1 Text and Figs Z and AA in S1 Text). Because it has a

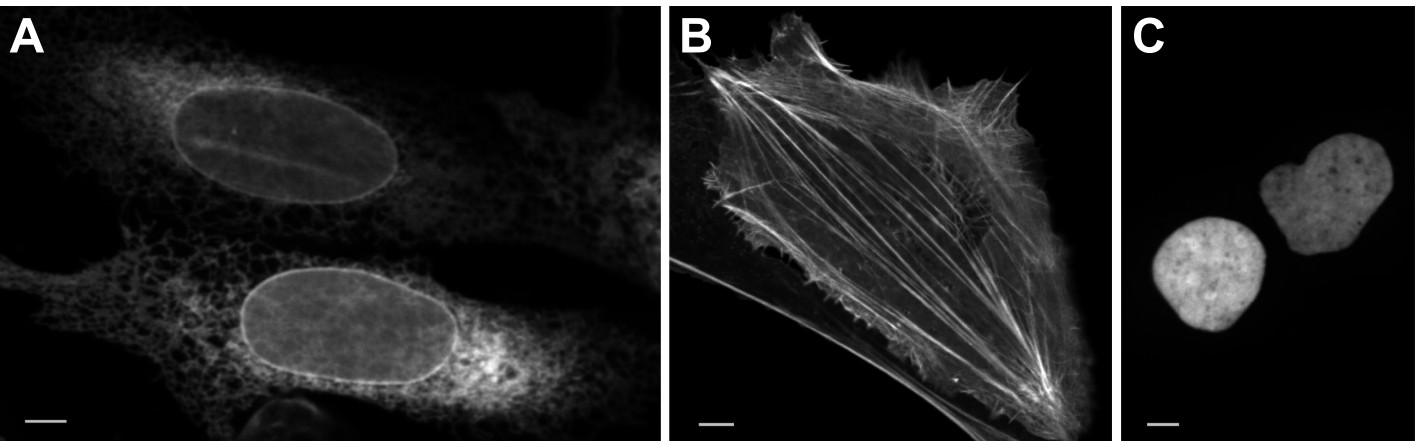

**Fig 4. Expression of mAvicFP1-tagged proteins in mammalian cells.** mAvicFP1 fusions to (A) CytERM, (B) LifeAct, and (C) H2B. U2-OS cells display expected localization. Scale bar is 10 mm. The data underlying this figure (raw image data) may be found at https://doi.org/10.26300/4x48-y393.

number of potentially useful properties, we consider AausFP1 the top candidate for future engineering among the FPs we have identified in this work.

Apart from AausFP1, an unexpected find among the newly discovered *A. victoria* FP homologs was AvicFP1, a transcript with relatively low abundance (Table A in S1 Text) but with high homology to avGFP (80% amino acid identity; see Fig A in S1 Text). At neutral pH, AvicFP1 has a single absorbance peak at 481 nm, indicating that its chromophore exists in a fully anionic state. It is curious that AvicFP1 would appear to be a superior energy transfer acceptor for the photoprotein aequorin than avGFP based on their absorbance spectra (Fig 2). However, avGFP was expressed at the sites of luminescence (bell margin), while AvicFP1 was only detected in the body of the animal (Table A in S1 Text), indicating that it is unlikely to be the natural energy acceptor for aequorin. Unfortunately, investigation of the interactions between AvicFP1 and aequorin are beyond the scope of this study.

The fluorescence $pK_a$ of AvicFP1 (4.9) is lower than that of EGFP (6.0). We speculate that the cysteine present in the first chromophore position is partly responsible for AvicFP1's EGFP-like properties. The S65T substitution in avGFP was among the most critical early mutations introduced to generate an all-anionic chromophore, though S65C was reported at the same time [2,16]. Because of mutations derived from errors in the oligonucleotides used for synthetic gene assembly, we also identified 1 colony among the thousands of initial AvicFP1 clones that produced a much larger proportion of mature FP in *E. coli* incubated at 37˚C. This clone contained a single point mutation leading to the substitution F64L, generating a variant with optical and biochemical properties indistinguishable from those of the wild-type protein. The F64L mutation is another of those originally identified for improving the folding of avGFP to ultimately produce EGFP [2,17].

Essentially all of the side chains that participate in the weak dimer interface of avGFP are conserved in AvicFP1. We hypothesized that mutations sufficient to monomerize avGFP variants (i.e., A206K [18]) would also produce a monomeric variant of AvicFP1. Using the organized smooth endoplasmic reticulum (OSER) assay to test for oligomeric behavior in cells [19], we found that the mutant AvicFP1-F64L/A206K displays monomeric behavior equivalent to mEGFP, widely considered the "gold standard" of monomeric FPs [19] (OSER data are summarized in Table B in S1 Text). Fusions to LifeAct [20] and histone 2B (H2B) displayed the expected localization and dynamics (Fig 5, S1 Movie and S2 Movie). Additionally, cells expressing H2B-mAvicFP1 and imaged at 2-minute intervals for 72 hours at 37˚C showed no

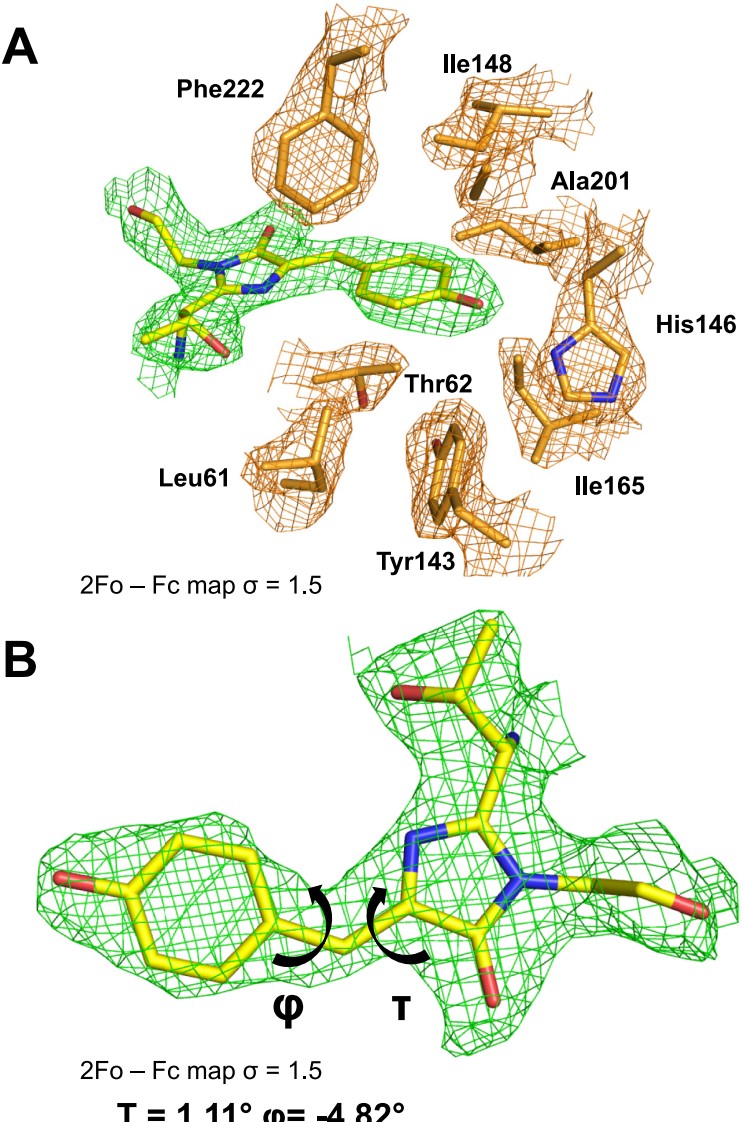

**Fig 5. The AausFP1 chromophore environment.** (A) $2F_{obs} - F_{calc}$ electron-density map contoured at a 1.5 σ level superimposed over the model of the chromophore and the neighboring residues in the structure of AausFP1. (B) Dihedral angle definition around the chromophore methylene bridge. The data underlying this figure may be found in PDB 6S67.

significant increase in doubling time (see Fig Y in S1 Text and S1 Data). We therefore decided that this variant merited an official name: mAvicFP1 (monomeric *A. victoria* fluorescent protein 1).

Originally, avGFP was identified as a partner to the photoprotein aequorin, and this association ultimately led to cloning the cDNA that encodes it. We speculate that other green-emitting FPs were not identified at the same time as avGFP because the brightest visible fluorescence in *A. victoria* is around the bell margin, while AvicFP1 appears to be expressed

exclusively in other tissues (Fig A in S1 Text). The optical properties of mAvicFP1 are superficially similar to those of mEGFP, and these FPs have similar brightness. In our hands, mAvicFP1's photostability under widefield and confocal illumination is somewhat higher than that of mEGFP (Figs Z and AA in S1 Text and S1 Data), its monomeric character is comparable, and its toxicity (as measured by the rate of cell division when expressing an H2B fusion; see S1 Text and Fig Y in S1 Text) appears to be lower that of mEGFP. However, the primary differentiating property of mAvicFP1 is its low $pK_a$, which may offer advantages when labeling proteins in acidic compartments. Others have also reported that mAvicFP1 spontaneously "blinks" under high illumination intensity, making it highly useful for single-molecule localization microscopy with a single excitation wavelength [21].

## Unusual *Aequorea* CPs

In the context of the broad phenotypic diversity of hydrozoan FP homologs now known [22–25], the presence of green- and red-absorbing CPs in *Aequorea* species is not surprising. However, the properties of *Aequorea* CPs differ in surprising ways from those previously cloned from other organisms. Every *Aequorea* CP displays a broad absorbance spectrum (Fig 2) that lacks the well-defined sharp peak and short-wavelength shoulder typical of most FPs and CPs, suggesting that these proteins contain an unusual chromophore and/or chromophore environment. Also, none of the *Aequorea* CPs has any measurable red fluorescence emission, even on our most sensitive instruments. All CPs described here migrate as high-molecular-weight, apparently soluble aggregates or high-order oligomers on a gel filtration column when expressed in *E. coli* (see Fig BB in S1 Text).

AausFP2 has a distinctive cyan-blue pigmented appearance when expressed in *E. coli*, with a broad absorbance spectrum peaking at 610 nm. *A.* cf. *australis* expresses a second CP, AausFP3, that displays a similarly symmetrical, shoulder-less absorbance peak, but with a maximum absorbance at 590 nm. X-ray crystallography analysis of AausFP2 (Tables B and C in S1 Text) revealed a conserved dimer interface geometry containing many conserved residues between AausFP1 and AausFP2. We suspect that despite AausFP2's behavior in gel filtration experiments, the native oligomeric state of AausFP2 may be a dimer. The amino acid residues making up the dimer interface in the AausFP2 crystal structure are also largely conserved across the other *Aequorea* CPs (Fig A in S1 Text), suggesting that if this is the native oligomeric state of AausFP2, then they are all likely to be dimers.

The X-ray crystal structure of AausFP2 further revealed a chemically novel chromophore in which the side chain of a neighboring cysteine is covalently linked to the methylene bridge of a twisted GFP-like chromophore (Fig 6; Tables D, E, and G in S1 Text; Figs F and H in S1 Text). This amino acid, Cys62, is conserved in all *Aequorea* CPs. The C62S mutant of AausFP2 appears yellow and has a major absorbance peak characteristic of a GFP-type chromophore (Fig I in S1 Text), strongly suggesting that this conserved cysteine is necessary for formation of the red-shifted chromophore. The peak absorbance wavelength of alkali-denatured *Aequorea* CPs displays a 20- to 30-nm red-shift relative to that expected for a GFP-type chromophore [2], which is abolished by addition of β-mercaptoethanol (Fig CC in S1 Text), providing additional evidence for the role of this unusual bond. Quantum mechanical calculations indicate that both the presence of a sulfur atom and a twisted chromophore are required to produce long-wavelength absorbance (see S1 Text, Fig J in S1 Text, and Table F in S1 Text).

Because of its broad absorbance reaching into the far-red and near-infrared regions of the spectrum, its relatively high extinction coefficient, and its efficient folding at 37°C, AausFP2 or its derivatives could ultimately prove very useful as photoacoustic tomography probes for deep tissue imaging.

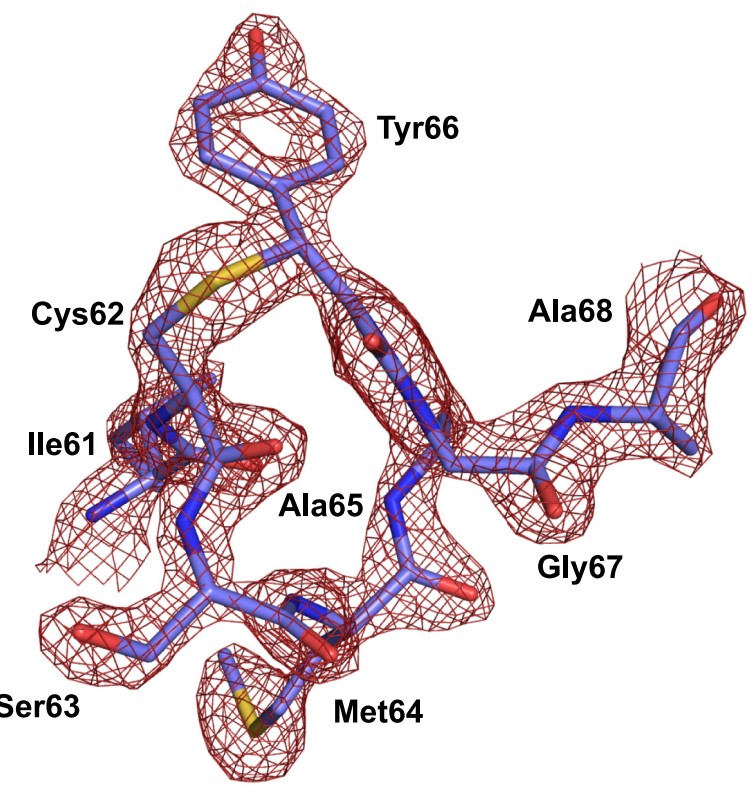

**Fig 6. $2F_{obs} - F_{calc}$ electron-density map contoured at a 2.0 σ level superimposed over the model of the chromophore and the 3 covalently bonded residues in the structure of AausFP2.** The data underlying this figure may be found in PDB 6S68.

Unlike their orthologs in *A*. cf. *australis*, which mature fully to their long-wavelength forms in the dark, the *A. victoria* CPs mature very slowly in the absence of blue light. When expressed in total darkness, AvicFP2 has peak absorbance in the blue region, and is weakly green fluorescent, suggesting an avGFP-type chromophore. Upon blue light exposure, AvicFP2 converts into a purple-blue CP with peak absorbance at 588 nm. In light of the quantum mechanical calculations presented (Fig J in S1 Text and Table F in S1 Text), this dramatic absorbance shift suggests that the light-induced change in AvicFP2 represents either the bonding of the Cys62 side chain to the methylene bridge of the chromophore or twisting of the chromophore from a planar to non-planar conformation.

AvicFP3 is highly homologous to AvicFP2 (96% amino acid identity; see Fig A in S1 Text), and is similarly green fluorescent when expressed and purified in the dark. Like AvicFP2, AvicFP3 converts to a green-absorbing CP when exposed to blue light, but appears to mature more efficiently than AvicFP2 in the absence of light (see pre-conversion absorbance spectrum; Fig 2). We anticipate that these proteins, if they can be engineered into monomers that mature efficiently at 37°C, could be useful as photoactivatable Förster resonance energy transfer (FRET) quenchers.

## A reversibly photochromic CP

The final FP homolog we identified in *A*. cf. *australis* is AausFP4, a very weakly fluorescent (quantum yield < 0.001) green-emitting CP with photochromic behavior strikingly similar to

that of the engineered avGFP variant Dreiklang [26]. When expressed and/or stored in the dark, AausFP4 reaches an equilibrium state with a major absorbance peak at 338 nm, indicating that the chromophore is neutral and missing at least 1 double bond relative to a mature GFP-type chromophore. With exposure to UV light, AausFP4 fully converts to an anionic GFP-like state with 477-nm peak absorbance.

This transformation is reversible by exposure to bright blue light or by storage in the dark. Together, these properties suggest a mechanism similar to that of Dreiklang, in which a structural water molecule can reversibly hydrate the imidazolinone ring of the chromophore via 2 different photochemical reactions triggered by different wavelengths of light [26]. A key difference between AausFP4 and Dreiklang is the absence of an approximately 400-nm absorbance peak in the "on" state, accompanied by off-switching mediated by blue rather than violet light. While AausFP4 is likely to be dimeric and/or aggregating like its closest relatives (AausFP2 and AausFP3), it may prove to be a useful starting material from which to engineer a new lineage of reversibly photoswitchable FPs or CPs. AausFP4 also likely represents, to our knowledge, the first naturally occurring example of Dreiklang-type photoswitching to be discovered.

## Conclusion

We have identified several new *Aequorea* FPs with the potential to further diversify the landscape of fluorescent probes and biosensors. AausFP1, the brightest fluorescent protein currently known, will serve as the parent of an entirely new lineage of super-bright FP variants. As a parallel scaffold to avGFP derivatives in many ways, mAvicFP1 may be quickly adaptable to existing probes and biosensors. AausFP4 is the first natural example of Dreiklang-type photochromism and may help generate other useful variations on this mechanism. Four highly unusual *Aequorea* CPs provide truly novel engineering opportunities, including generating new far-red-emitting FPs, improved dark FRET acceptors, and photoacoustic probes, among many other potential uses.

The discovery and understanding of these new fluorescent proteins in *Aequorea* were made possible through a highly collaborative and interdisciplinary approach involving field collection work, basic molecular biology, next-generation sequencing and bioinformatics, protein engineering, microscopy, X-ray crystallography, and phylogenetics. We are optimistic that more studies with this kind of holistic approach will help elucidate many of the mysteries still hiding in the natural world. In the time that has elapsed since Osamu Shimomura's first sampling of *A. victoria* in Friday Harbor, it has become clear that there is an urgent need to explore and understand as much of the molecular biodiversity that exists in the world as possible before many organisms go extinct or become too rare to sample.

## Materials and methods

### Chemicals and other reagents

Unless otherwise noted, bacterial growth medium components were purchased from Fisher Scientific, antibiotics were purchased from Gold Biotechnology, and other chemicals were purchased from Sigma-Aldrich.

### Sample collection and RNA extraction

A single specimen of *A.* cf. *australis* was collected near Heron Island (Queensland, Australia) and processed on-site at the Heron Island Research Station (University of Queensland) (see "Ethics statement" for permit and collection details). A single individual of *A. victoria* was obtained from the aquaculture collections of the Birch Aquarium at Scripps. The animals

being kept in the exhibit tank at this time were originally obtained from the Aquarium of the Pacific (Long Beach, CA), where they have been bred in captivity for many generations. Notably, the *A. victoria* jellies are fed a diet of crustaceans, and so any hydrozoan-like FP transcripts identified must come from the jelly itself rather than from contamination of the mRNA sequencing (mRNA-Seq) library with prey-derived mRNAs.

Live samples were photographed and then anaesthetized with MgCl$_2$ prior to being dissected. The bell margin, bell, and mouth were dissected separately, and total RNA was extracted using RNeasy Plus Mini Kit (Qiagen) following the manufacturer's instructions. For *A.* cf. *australis*, the purified samples were combined and dried in a GenTegra RNA tube for transport back to the US. For *A. victoria*, samples from the 3 body regions were kept separate.

### Next-generation sequencing

Total RNA samples were used as input to generate Illumina-compatible mRNA-Seq libraries at the Scripps Research Institute Next Generation Sequencing Core facility. Total RNA underwent polyA selection prior to Illumina TruSeq library prep. Libraries were run on 1 NextSeq flowcell and generated between 25 and 35 million 150-bp paired-end reads per sample.

Transcriptomes for individual samples as well as the aggregate *A. victoria* transcriptome were assembled using Trinity [27,28] either on a custom workstation in the lab or using the public Galaxy bioinformatics server [29]. Read mapping was performed using Bowtie 2 alignment [30] and RSEM [31] for cross-sample comparison. Additional details on transcript verification are included in S1 Text and Figs O–W in S1 Text.

### Species identification

The identity of *A.* cf. *australis* was established using phylogenetic analysis; see detailed methods and results in S1 Text, S1 Fig and S2 Fig. The identity of *A. victoria* was verified by the presence of an assembled transcript encoding avGFP, as well as its well-characterized morphology.

### Protein tree

Protein sequences were aligned with Clustal Omega [32], and a Bayesian tree was created using a burn-in of 3,000 iterations, run length of 30,000 iterations, and 4 chains with MrBayes software [33].

### Cloning and mutagenesis

Candidate FP-encoding transcripts were identified by BLAST homology searching using avGFP as the query against the assembled transcriptome databases as well as intermediate assembly files created by the Trinity workflow. Searching through intermediate assembly files allowed us to identify potential alternative transcript sequences and those that were (possibly incorrectly) collapsed into single contigs by Trinity. Putative FP-encoding transcripts were validated against raw read data and reconstructed as necessary (see below for detailed methods, results, and discussion). Sequence alignments were performed using Clustal Omega [32].

For each avGFP homolog identified, the coding region was identified and a synthetic gene was designed to produce the encoded polypeptide sequence using codons optimized for both human and *Escherichia coli* expression using an in-house BioXp 3200 instrument (SGI-DNA, La Jolla, CA) or ordered as a gBlock double-stranded gene fragment (Integrated DNA Technologies, San Diego, CA). Both PCR-amplified and synthetic cDNAs contained additional nucleotides at the 5′ end (GAAAACCTGTACTTCCAGGGT) and 3′ end

(CGTTTGATCCGGCTGC). Fragments encoding FPs were inserted using Gibson assembly [34] into the vector pNCST (modified from [35]) that had been PCR-amplified with the oligos pNCST-vec-F and pNCST-vec-R (Table H in S1 Text). The pNCST plasmid contains a synthetic promoter that drives high-level constitutive expression in most *E. coli* strains. This plasmid encodes an N-terminal 6xHis tag and linker followed by a TEV protease cleavage site just before the start codon of the inserted gene.

Site-directed mutagenesis of AvicFP1 was performed by generating 2 fragments of the FP coding sequence by standard PCR with Phusion polymerase (New England Biolabs) and primers as listed in Table H in S1 Text. Mutations were placed in the overlapping sequence between fragments to facilitate Gibson assembly of full-length mutant sequences in a 1-step insertion into the pNCST vector. Plasmids encoding AausFP1, mAvicFP1, and fusions of mAvicFP1 to H2B, LifeAct, and CytERM driven by a CMV promoter for mammalian expression were generated by Gibson assembly of the PCR-amplified FP sequence with the corresponding PCR-amplified fragment of pC1-mNeonGreen, pmNeonGreen-H2B-N-6, pmNeonGreen-LifeAct, and pmNeonGreen-CytERM [35].

## Recombinant protein purification

Sequence-verified plasmids were transformed into NEB5a strain *E. coli* (New England Biolabs) (because the promoter in the pNCST vector is semi-constitutive in most strains of *E. coli*, we find it convenient to use a single strain for cloning and expression), plated on LB/agar supplemented with carbenicillin (100 μg/ml), and incubated overnight at 37˚C. For proteins that matured efficiently at 37˚C (AvicFP-F64L, mAvicFP1, AvicFP4, AausFP1, AausFP2, EGFP, mEGFP, and mNeonGreen), colonies were picked and inoculated directly into a 200-ml baffled wide-mouth flask containing 50 ml of 2xYT broth and 100 μg/ml carbenicillin, and incubated overnight at 37˚C with shaking at 250 rpm. For proteins requiring multiple days at room temperature to mature (avGFP, AvicFP1, AvicFP2, AvicFP3, AausFP3, and AausFP4), a single colony was resuspended in 10 ml of 2xYT medium, and 100 μl of this suspension was plated on 5 100-mm petri dishes containing LB/agar and 100 μg/ml carbenicillin. After overnight incubation at 37˚C to initially establish colonies, plates were then incubated at room temperature for several days in the dark.

Bacteria containing the recombinant protein were recovered by centrifuging liquid cultures in 50-ml conical tubes at 4,500*g* for 10 minutes. For proteins expressed on LB/agar plates, a razor blade was gently glided over the surface of the agar, harvesting the colonies on the blade, and then wiped into 2-ml microcentrifuge tubes and gently centrifuged to the bottom of the tube. Four milliliters of the lysis reagent B-PER (Thermo 78248) was added for every gram of *E. coli* pellet. Tubes were gently vortexed until the pellets were completely dissolved, taking care not to form bubbles from the detergent component of the B-PER. The resulting suspension was then incubated on a gentle rocker for 15 minutes and then centrifuged at >20,000*g* for 10 minutes to pellet insoluble debris. Note that we find that there is a strong correlation between true protein solubility and extraction efficiency in B-PER that is not true of other extraction methods such as sonication, which can solubilize aggregated FPs more readily.

Meanwhile, we prepared a purification column by adding 1–2 ml of Ni-NTA resin slurry (Expedeon) into a 15-ml gravity column (Bio-Rad), allowing the storage buffer to drip through. The column was equilibrated with 10 bed volumes of wash buffer (150 mM Tris [pH 7.5], 300 mM NaCl, 5 mM imidazole) and then capped at the bottom. After centrifugation, the lysate was directly added to the prepared Ni-NTA column. The column was then capped at the top and the lysate-resin slurry was tumbled end-over-end for 30 minutes at 4˚C. The top/bottom caps were removed, and the liquid was allowed to drip through by gravity flow. The

column was then washed 3 times with 3 column volumes of wash buffer. Finally, the protein was eluted from the column by gradual addition of elution buffer (50 mM Tris [pH 7.5], 150 mM NaCl, 200 mM imidazole). Clear liquid was allowed to drip through, and only the fluorescent/colorful fraction was collected.

The proteins were then concentrated further using a 3-kD MWCO column (Amicon/Millipore) until the volume of protein solution was <150 μl. Meanwhile, 2× desalting columns (Pierce) were prepared for each protein by equilibrating in 50 mM Tris (pH 8.5)/150 mM NaCl according to the manufacturer's instructions. Then 150 μl of protein solution was loaded onto the equilibrated desalting column and centrifuged at 1,500 rpm for 1 minute in a microcentrifuge. The collected protein was then passed through a second equilibrated desalting column to ensure complete buffer exchange.

## Mammalian cell imaging

**Experiments performed in Dr. Shaner's lab.** U2-OS cells (HTB-96, ATCC) were grown in a 35-mm glass bottom dish (P35G-1.5-14-C, MatTek) or on coverslips (25CIRCLE #1.5, Fisherbrand) with DMEM (105666–016, Gibco) supplemented with 10% (v/v) FBS (10437–028, Gibco) under 5% humidified $CO_2$ atmosphere at 37˚C. Polyethylenimine (PEI) in double-distilled $H_2O$ (1 mg/ml (pH 7.3), 23966, Polysciences) was used as the transfection reagent. The transfection mixture was prepared in Opti-MEM (31985047, Thermo Fisher Scientific) with 4.5 μg of PEI and 500 ng of plasmid. For static images, a coverslip was placed in an Attofluor cell chamber (A7816, Invitrogen), and FluoroBrite DMEM (A18967-01, Gibco) was added. For time series, culture medium in glass bottom dishes was replaced with FluoroBrite DMEM (A18967-01, Gibco) supplemented with GlutaMAX (35050–061, Gibco) and 10% (v/v) FBS (10437–028, Gibco).

Confocal images and time series were acquired on a Leica TCS SP8 system using a 488-nm argon laser for excitation. The sample was placed in an incubation chamber with a controlled environment at 37˚C and 5% humidified $CO_2$ (Okolab). For LifeAct-mAvicFP1, a 63×/1.40 oil objective (HC PL APO CS2 63×/1.40 Oil, 15506350) was used with an emission bandwidth of 500–550 nm detected with a Leica HyD. For time series, a bandwidth of 500–600 nm was used, and images were acquired at 4.6-second intervals (4× line averaging, pinhole at 510 nm, 1 A. U.). For single images of H2B-mAvicFP1, CytERM-mAvicFP1, and CytERM-mEGFP (Addgene 62237), a 20× 0.75 NA air objective (HC PL APO 20×/0.75 CS2, 15506517) was used with an emission bandwidth of 500–550 nm detected with a HyD. For time series of these constructs, a bandwidth of 500–600 nm was used, and images were acquired at 3-minute intervals (4× line averaging, pinhole at 510 nm, 1 A.U.).

**Experiments performed at Harvard Medical School.** U2-OS cells were grown on #1.5 35-mm glass bottom dishes (MatTek) in McCoy's 5A medium supplemented with GlutaMAX (Thermo Fisher) and 10% fetal bovine serum (Thermo Fisher) and transfected with 0.5 μg of pCytERM-mAvicFP1 and pCytERM-mEGFP plasmid DNA using fuGENE (Promega) 24 hours prior to imaging. Before imaging, the growth medium was replaced with FluoroBrite DMEM supplemented with 5% FBS (Thermo Fisher). Image acquisition was performed in a full environmental enclosure (37˚C, 5% $CO_2$; Okolab) on a Nikon Ti-E microscope with Perfect Focus System, a Spectral Borealis-modified spinning disc confocal (Yokogawa X1), and an Orca Flash v3 sCMOS camera (Hamamatsu). OSER data were acquired with a 40× Plan Fluor 1.3 NA objective lens, and live time-lapse imaging was acquired with a 100× Plan Apo VC 1.4 NA objective (162-nm and 65-nm pixel size, respectively). Green fluorescence was excited with a 491-nm solid state laser (Cobolt) and a Di01-T405/488/568/647 (Semrock) dichroic; emission was selected with an ET525/50m filter (Chroma). Hardware was controlled with

MetaMorph (v7.8.13). For time-lapse experiments, single-plane images were acquired every second. For OSER acquisition, a uniform grid of images was acquired covering the entire coverslip.

Images were processed with Fiji [36]. OSER assay analysis was conducted as previously described [19].

## Cell division assay

U2-OS cells were grown and transfected as described above with plasmids encoding an N-terminal fusion of H2B to either mEGFP [18], AausFP1, or mAvicFP1, all with identical linker sequences. Prior to imaging, cells were stained with SiR-Hoechst (Cytoskeleton) following the manufacturer's instructions and maintaining a low concentration of stain in the medium throughout imaging. Cells were imaged on a Leica TCS SP8 system with an Okolab environmental chamber, as described above, starting 12–24 hours post-transfection. Images were collected every 2 minutes for >72 hours using 488-nm excitation with green emission to detect the H2B fusions, and with 633-nm excitation and far-red emission for the SiR-Hoechst stain to detect all DNA. For analysis, cells were selected from those expressing H2B and that underwent 1 cell division in the first half of the experiment. Control cells were selected from those neighboring the selected H2B-FP-expressing cells. The interval between cell divisions, defined as the time between visible chromosome separation, was recorded for the 2 daughter cells of each original cell.

## Photostability assay

U2-OS cells were grown and transfected as described above with plasmids encoding full-length untagged mEGFP, AausFP1, or mAvicFP1. Cells were imaged on a Leica SP8 laser scanning confocal microscope with a 63×/1.40 oil objective (HC PL APO CS2 63×/1.40 Oil, 15506350) at 37˚C and humidified 5% $CO_2$, as described above, with 488-nm argon laser illumination and an emission bandwidth of 500–700 nm detected with a photomultiplier tube. The pinhole was set to 2 A. U. at 510 nm to illuminate a thicker optical section of the cytoplasm at high intensity, at least partly accounting for diffusion of FP molecules in and out of the focal plane.

For widefield photobleaching, cells were imaged on a Nikon Ti-E microscope with a 40×/ 0.95 PL APO air objective, Perfect Focus System, a Spectra X light source (Lumencor) set to 470/24-nm bandpass excitation, 495-nm dichroic, 520/35-nm emission filter, and an Orca Flash v4 camera (Hamamatsu). Cells were imaged approximately 48 hours after transfection, with focusing using white light or very low power fluorescence excitation (≥1% imaging intensity for ≥5 seconds) to prevent pre-bleaching of the FPs.

The full-power light intensity at the sample plane was measured using a power meter (model 843-R, Newport), and the illumination spectrum at the objective was measured using a mini spectrometer fitted with a fiber optic input (Hamamatsu). For confocal bleaching, the light intensity measured at the objective was 250 μW. For widefield bleaching, the intensity at the objective was 10.3 mW. Cells were imaged for 8–10 minutes with continuous illumination (widefield) or continuous laser scanning (confocal). This was sufficient time to achieve >90% loss in fluorescence in most cases.

Image stacks were processed with Fiji [36], first by registering them with the StackReg plugin [37] to eliminate any artifacts caused by drift. A region of interest (ROI) was defined in the cytoplasm of each cell as well as a background region. We then measured the mean intensity of each ROI over the image stack and interpolated to determine the time value corresponding to 50% of the initial fluorescence signal for each cell. Photobleaching half-times were then scaled by a correction factor that corresponds to the per-molecule brightness of each FP under the specific illumination condition. For confocal bleaching, the correction factor

corresponds to the molar extinction coefficient at 488 nm. For widefield bleaching, the correction factor depends on both the absorbance spectrum of the FP and the illumination spectrum at the objective:

$$\phi \int I(\lambda) \cdot A(\lambda) d\lambda$$

where $\phi$ is the quantum yield, $I(\lambda)$ is the illumination intensity, and $A(\lambda)$ is the absorbance of the FP. In both cases, the correction factor normalizes the photobleaching half-times to those that would be observed if the excitation were tuned to produce equal photon output per FP molecule at time 0. These experiments and the analysis of the resulting data are discussed in more detail in S1 Text.

## Quantum yield and extinction coefficient determination

Purified green-emitting FPs were characterized as previously described [38] to determine quantum yield. Briefly, FPs that had been buffer-exchanged into 50 mM Tris-HCl (pH 8.5)/ 150 mM NaCl were diluted into the same buffer until the baselined peak absorbance was ≤0.05 as measured by a UV-2700 UV-Vis spectrophotometer (Shimadzu). Sample and standard (fluorescein in 0.1 M NaOH, quantum yield 0.95 [39]) absorbance were matched within 10% at 480 nm, the excitation wavelength used for fluorescence emission spectra. Immediately after measuring the absorbance spectrum, the cuvette containing the sample was transferred to a Fluorolog-3 fluorimeter (Jobin Yvon), and the emission spectrum was taken from 460 nm to 700 nm in 1-nm steps, with excitation at 480 nm and a slit width of 2 nm for both excitation and emission. Emission spectra were interpolated under the region in which scattered excitation light bleeds through into the emission path. Quantum yield was calculated by dividing the area under the sample emission curve by its absorbance at 480 nm and dividing by the same ratio for the standard, then multiplying by 0.95, the quantum yield of the standard.

Extinction coefficients for all FPs and CPs in this study were measured using the accepted standard method of measuring FP extinction coefficients, with alkali denaturation (addition of 2 M NaOH to the FP sample to a final concentration of 1 M NaOH) as previously described [38]. This method relies on the denatured chromophore absorbance and extinction coefficient to be invariant between FPs with chemically identical chromophores, and allows calculation of the extinction coefficient of the natively folded protein by comparing the peak height between native and denatured absorbance spectra. We performed this assay with the following modifications: (1) In order to avoid calculating erroneously large values of FP extinction coefficients from alkali denaturation measurements, several absorbance spectra were taken for each sample. Beginning immediately after addition of NaOH, multiple absorbance spectra were taken over several minutes to determine both the point at which the protein was fully denatured and the point at which it reached maximum absorbance at approximately 447 nm. The *maximum* measured value of the peak absorbance of fully denatured protein was used in extinction coefficient calculations. (2) For CPs containing the novel cysteine-linked chromophore, the peak absorbance of alkali-denatured protein is red-shifted 20–30 nm relative to the known 447-nm peak of GFP-type chromophores [2]. Because the extinction coefficient of this species is unknown, we also measured absorbance spectra for alkali-denatured CPs with the addition of 1 mM β-mercaptoethanol, which is expected to break the bond between the sulfur atom of the cysteine side chain and the methylene bridge of the chromophore, producing a GFP-type denatured chromophore. The maximum absorbance value of reduced, denatured chromophore was used in calculation of the extinction coefficient, which should be considered an estimate for *Aequorea* CPs pending much deeper investigation into the biochemical properties of their unique chromophore.

## pK$_a$ determination

Purified proteins were concentrated and desalted as described above into 20 mM Tris-HCl (pH 8). A solution of 50 mM Tris-HCl, 50 mM citric acid, 50 mM glycine, and 150 mM NaCl (final concentrations after pH adjustment) was prepared and split into 2 master stocks that were adjusted to pH 3 and pH 12 with HCl and NaOH, respectively. These stocks were then used to prepare buffers at pH 3, 4, 5, 6, 6.5, 7, 7.25, 7.5, 7.75, 8, 9, 10, 11, and 12 by mixing at different ratios. Each sample was then diluted (2 μl of sample + 198 μl of buffer) into each pH buffer, and its emission or absorbance was measured using an Infinite M1000 PRO (Tecan) plate reader. The pK$_a$ was determined by interpolating the pH value at which the fluorescence or absorbance value was 50% of its maximum.

## Size-exclusion chromatography and light scattering

Two milligrams of purified protein in 100 ul of running buffer was applied to a Shodex KW-802.5 column with guard column KW-G 6B (Showa Denko America, New York, NY) and run in 50 mM Na-HEPES/150 mM NaCl (pH 7.35) at a flow rate of 0.5 ml/minute using an Agilent 1100 Series HPLC system controlled by ChemStation software (Agilent Technologies, Santa Clara, CA). Protein elution was dually monitored with 280-nm absorbance and at the absorbance maxima for each fluorescent protein. In-line light scattering was performed by a Wyatt Heleos system running ASTRA software (Wyatt Technology, Goleta, CA). Clinical-grade cetuximab used as a molecular weight standard was obtained from the UCSD Moores Cancer Center pharmacy.

## Protein crystallogenesis

AausFP1 and AausFP2 were first expressed and purified as aforementioned. The His-tag was cleaved off using either TEV for AausFP1 (1/100 protease/protein ratio, overnight incubation at room temperature) or proteinase K for AausFP2 (1/50 protease/protein ratio, 1-hour incubation at room temperature). The protein solution was run through an additional His-Trap column to remove cleaved tag and uncleaved protein. A final purification step consisted of a gel filtration column (Superdex 75-10/300 GL, GE Healthcare, Chicago, IL). Fractions were analyzed using 15% SDS-PAGE gels, pooled and concentrated to 40 and 51 mg/ml for AausFP1 and AausFP2, respectively, using an Amicon Ultra centrifugal filter with a molecular weight cutoff of 30 kDa (Merck, Darmstadt, Germany). Initial crystallization hits were obtained using the HTX lab platform of the EMBL Grenoble Outstation, and then manually optimized. AausFP1 was crystallized with the hanging drop method using 0.7–1.3 M trisodium citrate, 0.2 M sodium chloride in 0.1M Tris buffer (pH 6.5–8.0). AausFP2 was crystallized with the hanging drop method using 14%–24% PEG 3350 trisodium citrate and 0.2 M sodium chloride in 0.1 M HEPES buffer (pH 7.3–8.2).

## Diffraction data collection

Diffraction data for AausFP1 were collected on beamline BL13-XALOC at the ALBA synchrotron in Barcelona (Spain) [40] from a crystal flash-cooled at 100 K in its mother liquor supplemented with 20% (v/v) glycerol for cryoprotection. Diffraction data for AausFP2 were collected on beamline ID30B of the European Synchrotron Radiation Facility in Grenoble (France) [41] from a crystal flash-cooled at 100 K without addition of any cryoprotectant. Diffraction data were integrated and reduced using XDS and XSCALE [42]. Data collection and reduction statistics are given in Table C in S1 Text.

## Structure determination

A BLAST search (https://blast.ncbi.nlm.nih.gov/) identified the fluorescent protein phiYFPv from the jellyfish genus *Phialidium* as the closest homolog of both AausFP1 and AausFP2 (sequence identities of 61% and 50%, respectively) with a known structure (PDB entry code 4HE4 [43]). The structures of AausFP1 and AausFP2 were solved by the molecular replacement method using the 4HE4 coordinates as a search model with the program PHASER [44]. The model was progressively and interactively modified in COOT [45] and refined with REFMAC5 [46]. The asymmetrical units contain 4 molecules for AausFP1 and 1 molecule for AausFP2. Analysis of the interaction interfaces with PISA [47] strongly suggests that the AausFP1 tetramer consists of a dimer of a physiological dimer (interface areas of 1,210 Å$^2$ versus 360 Å$^2$ for the third and fourth largest areas), while the AausFP2 monomer forms a physiological dimer with a symmetry-related molecule (interface area of 1,290 Å$^2$ versus 540 Å$^2$ for the second largest area). Structure refinement statistics are given in Table C in S1 Text.

## Calculation of AausFP2 absorption maxima

Eight models of the minimal part of the chromophore were constructed, modeling only the 2 conjugated cycles of the chromophore. H atoms replaced in all models the 2 alpha carbon atoms linking the chromophore to the rest of the protein. 3D coordinates for all heavy atoms of the chromophore were taken from the crystallographic structures without optimization, leading to 2 groups of models, one with the conformation of the EGFP structure and one with the conformation of the AausFP2 structure. The corresponding sets of models were labeled EGFP and AausFP2. The main difference between the 2 sets of models is the dihedral angle between the 2 cycles, i.e., −2° (almost planar) for EGFP and −53° (twisted) for AausFP2.

In each set of models, the phenol moiety was presented in its protonated form (neutral chromophore) or phenolate form (anionic chromophore). Moreover, in the AausFP2 set, the carbon between the 2 cycles of chromophore is linked to a protein's cysteine through a thioether bond, whereas this carbon is simply protonated in the case of EGFP. Therefore, in the models, this carbon was linked either to a mercapto group (–SH) or simply protonated. Structures were protonated and the position of H atoms were optimized at the B3LYP/6-31+g (d,p) level of theory with the Gaussian G09 program.

## Software

The web-based data-plotting software PlotsOfData [48] was used to generate Figs Y, Z, and AA in S1 Text.

## Ethics statement

All scientific collection in the field was performed under permit G17/39943.1 granted to Dr. Anya Salih, Western Sydney University, by the Great Barrier Reef Marine Park Authority. A specimen of *A.* cf. *australis* was collected within the Scientific Research Zone surrounding Heron Island (Queensland, Australia) using a hand-held net and was transported back to the lab in seawater. Live samples were kept in fresh running seawater for minimal amounts of time after collection.

## Supporting information

**S1 Data. Raw cell division and photobleaching data and corresponding analysis for Fig Y, Z, and AA in S1 Text.**
(XLSX)

**S1 Fig. 16S phylogenetic tree.** The 16S tree is inconclusive as to the phylogenetic position of both the transcriptomic 16S sequences and the reference-guided assembly 16S sequence. Several species are monophyletic in this tree and *A. australis* is in a large polytomy. See S1 Text for additional discussion.
(PDF)

**S2 Fig. COI phylogenetic tree.** The COI tree shows that the reference-corrected COI sequence (sample_COI) is sister to a large *A. australis* clade. See S1 Text for additional discussion.
(PDF)

**S1 Movie. Confocal imaging of H2B-mAvicFP1 expressed in U2-OS cells.** Scale bar is 10 mm. Timestamp is in hours:minutes. Images were taken at 3-minute intervals. Video playback is at 14 frames per second (total imaging duration 3 hours 9 minutes). Raw imaging data used to generate this movie are available at https://doi.org/10.26300/4x48-y393.
(MOV)

**S2 Movie. Confocal imaging of LifeAct-mAvicFP1 expressed in U2-OS cells.** Scale bar is 5 mm. Timestamp is in hours:minutes. Images were taken at 4.6-second intervals. Video playback is at 100 frames per second (total imaging duration 4 hours 35 minutes). Raw imaging data used to generate this movie are available at https://doi.org/10.26300/4x48-y393.
(MOV)

**S1 Text. Supporting materials and methods, results, and discussion.**
(PDF)

# Acknowledgments

We dedicate this manuscript to the memory of Dr. Roger Y. Tsien and Dr. Osamu Shimomura, whose studies on *A. victoria* and avGFP continue to inspire us and to catalyze new technologies for biological imaging. This work was also made possible by the Crystal Jelly exhibit at the Birch Aquarium at Scripps, highlighting the significance of this species in the history of biomedical research. This exhibit was the source of the *A. victoria* individual used in this work. The European Synchrotron Radiation Facility is acknowledged for access to beamline ID30B and facilities for molecular biology via its in-house research program. The ALBA synchrotron is acknowledged for allocation of beamtime on beamline BL13-XALOC. We thank Franck Borel, David Cobessi, and the beamline staff for help during data collection on BL13-XALOC. We also wish to thank Dr. Lauren M. Barnett for aiding in the collection of *A.* cf. *australis*, Wyatt Patry (Monterey Bay Aquarium) for helping in species identification, and Dr. Ute Hochgeschwender, Dr. Thomas Blacker, and Dr. Robert E. Campbell for helpful feedback on the manuscript.

# Author Contributions

**Conceptualization:** Gerard G. Lambert, Isabelle Navizet, Antoine Royant, Nathan C. Shaner.

**Data curation:** Gerard G. Lambert, Hadrien Depernet, Guillaume Gotthard, Darrin T. Schultz, Talley Lambert, Daphne S. Bindels, Antoine Royant, Nathan C. Shaner.

**Formal analysis:** Gerard G. Lambert, Hadrien Depernet, Darrin T. Schultz, Isabelle Navizet, Daphne S. Bindels, Antoine Royant, Nathan C. Shaner.

**Funding acquisition:** Nathan C. Shaner.

**Investigation:** Gerard G. Lambert, Hadrien Depernet, Guillaume Gotthard, Darrin T. Schultz, Isabelle Navizet, Talley Lambert, Stephen R. Adams, Albertina Torreblanca-Zanca, Meihua Chu, Anya Salih, Antoine Royant, Nathan C. Shaner.

**Methodology:** Hadrien Depernet, Guillaume Gotthard, Darrin T. Schultz, Isabelle Navizet, Talley Lambert, Stephen R. Adams, Albertina Torreblanca-Zanca, Meihua Chu, Daphne S. Bindels, Antoine Royant, Nathan C. Shaner.

**Project administration:** Nathan C. Shaner.

**Resources:** Darrin T. Schultz, Talley Lambert, Vincent Levesque, Jennifer Nero Moffatt.

**Software:** Isabelle Navizet.

**Supervision:** Antoine Royant, Nathan C. Shaner.

**Validation:** Darrin T. Schultz.

**Writing – original draft:** Gerard G. Lambert, Hadrien Depernet, Guillaume Gotthard, Darrin T. Schultz, Isabelle Navizet, Talley Lambert, Daphne S. Bindels, Antoine Royant, Nathan C. Shaner.

**Writing – review & editing:** Gerard G. Lambert, Hadrien Depernet, Darrin T. Schultz, Talley Lambert, Jennifer Nero Moffatt, Anya Salih, Antoine Royant, Nathan C. Shaner.

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
