## [Editor Report · Decision Letter 0]

8 Nov 2019

Dear Dr Shaner, 

Thank you for submitting your manuscript entitled "Aequorea victoria's secrets: diverse and unusual fluorescent protein genes beyond GFP" for consideration as a Research Article by PLOS Biology.

Your manuscript has now been evaluated by the PLOS Biology editorial staff, as well as by an academic editor with relevant expertise, and I'm writing to let you know that we would like to send your submission out for external peer review. IMPORTANT: We note that you've submitted this as a regular Research Article, but we feel it would be better considered as a Methods and Resources paper. No re-formatting is required at this stage, but please select "Methods and Resources" as the article type when you submit your remaining metadata (see next paragraph).

Please re-submit your manuscript within two working days, i.e. by Nov 11 2019 11:59PM.

Kind regards,

Roli Roberts

Senior Editor

PLOS Biology

---

## [Decision Letter · Decision Letter 1]

8 Jan 2020

Dear Dr Shaner,

Thank you very much for submitting your manuscript "Aequorea victoria's secrets: diverse and unusual fluorescent protein genes beyond GFP" for consideration as a Methods and Resources paper at PLOS Biology. Your manuscript has been evaluated by the PLOS Biology editors, an Academic Editor with relevant expertise, and by four independent reviewers. Please accept our apologies for the delay incurred over the holiday period.

IMPORTANT: You'll see that while reviewers #1, #2 and #3 are broadly positive about your study, they each make further requests for improvements. Reviewer #4 is less positive, but may not have been considering this paper as a candidate for our Methods and Resources section; however, after discussion with the Academic Editor, we ask that you do address the concerns and requests raised by reviewer #4. We would also echo reviewer #1's concerns about the unusual organisation of your manuscript, which we think our readers will find very unclear; please rectify this.

In light of the reviews (below), we will not be able to accept the current version of the manuscript, but we would welcome re-submission of a much-revised version that takes into account all of the reviewers' comments. We cannot make any decision about publication until we have seen the revised manuscript and your response to the reviewers' comments. Your revised manuscript is also likely to be sent for further evaluation by the reviewers.

We expect to receive your revised manuscript within 2 months. 

**IMPORTANT - SUBMITTING YOUR REVISION**

*NOTE: In your point by point response to to the reviewers, please provide the full context of each review. Do not selectively quote paragraphs or sentences to reply to. The entire set of reviewer comments should be present in full and each specific point should be responded to individually, point by point.

*Re-submission Checklist*

*Published Peer Review*

*PLOS Data Policy*

*Blot and Gel Data Policy*

Sincerely,

Roli Roberts

Senior Editor

PLOS Biology

REVIEWERS' COMMENTS:

Reviewer #1:

The manuscript by Lambert et al. notifies us of nature's profundity. The Aequorea victoria jelly fish species, which provided us with the avGFP, has proven to possess some additional novel fluorescent proteins (FPs). I agree to the last paragraph of the main text in that we now need to explore and understand as much of the molecular biodiversity of glowing creatures. 

I found the manuscript somehow disorganized. It was difficult to follow the logic in the Introduction, Results and Discussion. I'd recommend that the manuscript be rewritten more carefully. 

My comments are as follows.

1) None of supplementary data (figures and tables) are referred to in the main text. Also, the Introduction contains some results; Figures 2 and 3 are already referred to there. 

2) Page 2, right, line 5. Please define "AausFP." It should be "A. cf. australis FP." 

3) Figure 2. What were the concentrations of the protein samples?

4) Page 2, right, lines 19-. The low pKa of AvicFP1, which contains CYG, is explained by analogy to the S65C mutant of avGFP. I think, however, this portion is too speculative. 

5) Page 3, left, line 27. I think that doubling time of H2B-mAvicFP1 should be provided.

6) Page 3. "It is somewhat ironic that,,," avGFP absorbs principally violet light (400 nm) for the green fluorescence whereas AvicFP1 absorbs blue light. I would rather discuss why avGFP is abundant in A. victoria. 

7) Why aren't the two CPs from A. cf. australis called AausCP2 and AausCP3?

8) Figure 3. The dotted lines for unconverted spectra are very hard to see.

9) Figure 4. Hard to see. I would recommend that a relatively simple phylogenic tree be put. 

10) Page 4, left, lines 10-. Why are the tandem-dimers and monomers hidden?

11) Page 5, right. About the photoconversion of AvicFP2 and AvicFP3. Could the authors identify the blue-absorption peaks for photoconversion in Figure 1? Also, I am negative on their "rapid" photoconversion, considering that It took seconds to minutes. Ideally, the authors should measure the quantum efficiency of the conversion.

12) There is no information of photostability of the new FPs. To show their practical usefulness, photobleaching data must be presented. 

Reviewer #2:

The work presented in this excellent manuscript involved a collaborative team finding and analysing a new suite of fluorescent proteins from the original Aequorea victoria and a related Aequorea species. The group identified a number of new, interesting and potentially useful fluorescent proteins. Two of these were further characterized by structural biology to provide interesting insights into the molecular basis of function. I guess the most interest going forward is AausFP1 in terms of its overall brightness (which is authors has eluded too), but I am sure there is a lot to be learned (for both fundamental and long-term application) from the others. 

Overall, I thought this was an excellent piece of work with very little to argue with - the science was excellent and I enjoyed reading the manuscript. It is clear that this work is publishable in PLoS Biology as it stands. My reason for minor revisions is that I suggest a couple of changes (and they are minor in the scheme of things).

1. Oligomerisation can play an important role in the spectral properties of fluorescent proteins. As I read it, apart from the engineered mAvicFP1, all the other proteins were oligomers. Do the authors have other data than OSER to confirm the oligomeric status of the proteins (e.g. size exclusion, dynamic light scattering)? Some information is inferred from the two crystal structures but such data can sometimes be a little misleading in terms of oligomeric state in solution. It might also be worth adding a column to Table 1 to state the oligomeric form of each fluorescent protein. 

2. Leading on from point 1, it is known that oligomerisation of normally monomeric fluorescent proteins can impact of the spectral characteristics. AausFP1 may be a good example (although given the comments in the manuscript, there is more to be told on this at a later date). A recent good example is DOI: 10.1038/s42004-019-0185-5 concerning how oligomerisation can have a positive impact on GFP properties. It might be good to include such information.

3. Very little is said about maturation times expect that in the methods it appears that certain variants matured at different rates. Could the authors add which proteins matured under which particular conditions. 

4. While I don't dispute the extinction coefficients measured, I was surprised to see that the proteins where denatured to calculate them. Did the authors look at the values for the native proteins?

Reviewer #3:

This is a very valuable resource to the scientific community in that it identifies a number of new sequences for fluorescent proteins and chromoproteins and characterizes the basic properties of those proteins. These sequences increase the diversity of naturally derived sequences and could be useful for incorporating into efforts aimed at gene shuffling. The paper is clearly written and the resource and results are logically presented. I don't find any flaws with the analysis or presentation. The only thing I would request is that the axes in Figure 3 and the size of Figure 4 be increased because the text isn't legible. Otherwise I think this paper makes a valuable contribution to the scientific community and I support its publication. 

Reviewer #4: 

Lambert G.G. et al. cloned homolog genes of the Aequorea Victoria GFP from Aequorea sp. by mRNA-Seq and de novo transcriptome assembly. Isolated fluorescent proteins (FPs) and chromo proteins (CPs) were characterized, and distinct features such as a chromophore with a crosslink to the main polypeptide chain and reversibly photoswitchability were identified.

To be honest, I got an impression to this manuscript such that the principal claim of the authors is unclear and it seems like a catalog of the FPs and CPs released from the author's group. To discuss the molecular evolution, meticulous comparison with FP/CPs derived from multiple species in the Aequorea sp. is essential. Meanwhile, for claiming the applicability of FP/CPs as a valuable resource for bioimaging, availability to the practical imaging technique using their distinct features must be shown. However, the authors did not seriously address these issues. Therefore, it is difficult to be accepted as a paper worthy to publish from this journal. I recommend for authors to submit this manuscript to much more specific journal. 

Minor points:

1. The descRiption in the Introduction section is not match with the contents in the Results and Discussion section.

2. There are many Supplementary Materials which are not cited in the text.

3. Fig. S12 and Fig. S-A to I in Supplementary Materials are unclear to make out the letters in the figures.

---

## [Decision Letter · Decision Letter 2]

16 Sep 2020

Dear Dr Shaner,

Thank you for submitting your revised Methods and Resources entitled "Aequorea victoria's secrets: diverse and unusual fluorescent protein genes beyond GFP" for publication in PLOS Biology. I've now obtained advice from one of the original reviewers and have discussed their comments with the Academic Editor. 

Based on the reviews, we will probably accept this manuscript for publication, assuming that you will modify the manuscript to address the remaining points raised by the reviewer. 

IMPORTANT:

a) As well as the remaining points from reviewer #1, please could you attend to my Data Policy requests further down this email?

b) I'm afraid that after some discussion, we think that your title is not appropriate or informative enough. Specifically, while we appreciate the neat pun, we think that the allusion to a company whose products could be seen by many readers as objectifying women is not appropriate. Maybe something like "New jellyfish fluorescent proteins with unique properties for bioimaging and biosensing"?

We expect to receive your revised manuscript within two weeks. Your revisions should address the specific points made by each reviewer. In addition to the remaining revisions and before we will be able to formally accept your manuscript and consider it "in press", we also need to ensure that your article conforms to our guidelines. A member of our team will be in touch shortly with a set of requests. As we can't proceed until these requirements are met, your swift response will help prevent delays to publication.

- a cover letter that should detail your responses to any editorial requests, if applicable

*Copyediting*

*Published Peer Review History*

*Early Version*

Sincerely,

Roli Roberts

Senior Editor,

rroberts@plos.org,

PLOS Biology

DATA POLICY:

We note that many of your Fig panels are photos, or present structural, phylogenetic, sequence or spectral data, all of which are available in the supplement or in repositories. However, we also need you to provide the underlying data for Figs S14, S15, S16. NOTE: the numerical data provided should include all replicates AND the way in which the plotted mean and errors were derived (it should not present only the mean/average values).

Please also ensure that figure legends in your manuscript include information on where the underlying data can be found (including repositories), and ensure your supplemental data file/s has a legend.

REVIEWERS' COMMENTS:

Reviewer #1:

This is an omnibus study that covers fluorescent proteins and chromoproteins from Aequorea species, and I agree that AausFP1 would be the top candidate for future engineering because of its extreme brightness. I think that the revised manuscript should be almost publishable. I have two minor criticisms as follows. 

I was interested in the observation that AvicFP1 was localized in the body and mouth but not the bell margin where the Aequorin-based bioluminescence occurs. 

Page 3, right. Lines 18-

"We can only speculate about what biological imaging might look like today if AvicFP1, rather than avGFP, had been the first FP cloned."

This is a vague sentence. I would suggest that the authors refer to the fact that avGFP was originally discovered as a protein accompanying Aequorin. 

Page 4, left. 

"To our knowledge, there have been no previous reports of chromoproteins with a quantum yield of absolutely zero." 

Please specify quantum yield values of AausFP2 and AausFP3 compared to others.

---

## [Editor Report · Decision Letter 3]

15 Oct 2020

Dear Dr Shaner,

On behalf of my colleagues and the Academic Editor, Ana J Garcia-Saez, I am pleased to inform you that we will be delighted to publish your Methods and Resources in PLOS Biology. 

Early Version

PRESS 

Kind regards,

Erin O'Loughlin

Publishing Editor, 

PLOS Biology

on behalf of

Roland Roberts,

Senior Editor

PLOS Biology